# Orthotopic T-Cell Receptor Replacement—An “Enabler” for TCR-Based Therapies

**DOI:** 10.3390/cells9061367

**Published:** 2020-06-01

**Authors:** Kilian Schober, Thomas R. Müller, Dirk H. Busch

**Affiliations:** 1Institute for Medical Microbiology, Immunology and Hygiene, Technische Universität München (TUM), 81675 Munich, Germany; thm.mueller@tum.de; 2German Center for Infection Research (DZIF), Partner Site Munich, 81675 Munich, Germany

**Keywords:** TCR engineering, CRISPR/Cas9, orthotopic TCR replacement, OTR, gene editing, adoptive cell therapy, immunotherapy, T-cell therapy, gene therapy

## Abstract

Natural adaptive immunity co-evolved with pathogens over millions of years, and adoptive transfer of non-engineered T cells to fight infections or cancer so far exhibits an exceptionally safe and functional therapeutic profile in clinical trials. However, the personalized nature of therapies using virus-specific T cells, donor lymphocyte infusion, or tumor-infiltrating lymphocytes makes implementation in routine clinical care difficult. In principle, genetic engineering can be used to make T-cell therapies more broadly applicable, but so far it significantly alters the physiology of cells. We recently demonstrated that orthotopic T-cell receptor (TCR) replacement (OTR) by clustered regularly interspaced short palindromic repeats (CRISPR)/ CRISPR-associated protein 9 (Cas9) can be used to generate engineered T cells with preservation of near-physiological function. In this review, we present the current status of OTR technology development and discuss its potential for TCR-based therapies. By providing the means to combine the therapeutic efficacy and safety profile of physiological T cells with the versatility of cell engineering, OTR can serve as an “enabler” for TCR-based therapies.

## 1. Introduction

T cells are fundamentally important for the immune system in order to control viral infections and tumors [1,2]. This insight forms the basis for the concept of T-cell therapies, which are revolutionizing medicine. While checkpoint inhibition [3] and T cells engineered to express chimeric antigen receptor (CAR T cells) [4] represent two of the most successful approaches, they simultaneously epitomize two poles in the spectrum of immunotherapies. Checkpoint inhibitors re-activate polyclonal endogenous (physiological) T cells. This circumvents the need for tedious engineering and has the advantage of exploiting the potential of T cells with different antigen specificities and affinities. However, this also means that therapy success is often not predictable, and side effects due to the stimulation of auto-reactive T cells can be severe.

Adoptive transfer of non-engineered T cells addresses some of these problems in the form of donor-lymphocyte infusions (DLI) [5], tumor-infiltrating lymphocytes (TIL) [6], or pathogen-specific T cells [7], which have been applied for more than three decades with excellent safety and promising efficacy profiles. Selecting for antigen specificity can also render non-engineered T-cell products more defined [8,9]. However, broad applicability of non-engineered T cells is still a limiting factor since suitable donors (with antigen-specific memory populations and matched human leukocyte antigens (HLA)) have to be available.

Engineering T cells to express a transgenic antigen-specific receptor can address the problem of suitable donors since autologous T cells can be used. The first proof of concept to re-direct T cells via a T-cell receptor (TCR) [10] or CAR [11] was demonstrated in the 1980s. To date, anti-cluster of differentiation 19 (CD19) CAR T cells represent the poster child of immunotherapy [12]. CAR T cells target a specific antigen and, therefore, represent a defined therapeutic product. However, current CAR T-cell manufacturing protocols are still lengthy and cumbersome, which can introduce considerable heterogeneity into the “living drug”. In clinical reality, the success rate and the speed of generating cell products are some of the most important bottlenecks of applying CAR T cells. Furthermore, it is questionable how much of the success of CAR T cells can be explained by the nature of the CAR itself. Instead, it is arguably the target ligand CD19 that makes therapy with anti-CD19 CAR T cells so prosperous. B-cell malignancies often homogeneously express CD19 as a lineage marker on the cell surface, and aplasia of healthy B cells is for some time acceptable or can be counteracted upon. For most other tumors, such an exceptional target simply does not exist [12].

In contrast to CAR T cells, TCR engineered T cells are not limited to target surface antigens, and important studies demonstrated the feasibility of using TCR-engineered T cells for therapy of different cancer entities [13,14,15,16,17]. While these first studies targeted overexpressed tumor antigens, T cells that react to checkpoint therapy often recognize mutation-derived neo-epitopes [18]. Neo-epitopes also bear the advantage of being very specific to the tumor itself, ameliorating the danger of off-target recognition [18]. Importantly, targeting neo-epitopes can only be achieved via the TCR (and not, for example, with a CAR).

Overall, TCR engineered T cells, therefore, have enormous potential as therapeutic agents, but TCR engineering entails problems of its own. Conventionally, T cells are equipped with a transgenic TCR via viral transduction [10,19]. However, this way of engineering entails several challenges. Firstly, the endogenous TCR remains. This leads to diminished functionality of transgenic TCRs through suppressed surface expression [20] and the generation of putative mispaired TCR variants, which can cause auto-reactivity or graft-versus-host disease (GvHD) [21,22]. Secondly, viral transduction leads to random (or at least semi-random) integration of the transgene. This poses a safety concern, and it also makes extrinsic gene promotors necessary that guarantee transgene expression from unknown genomic loci. In contrast to non-engineered T cells, which downregulate the TCR after antigen contact [23,24], TCR transduced T cells hardly downregulate the TCR [25,26], indicating artificial transgene regulation that could stem from the fact that extrinsic promotors drive transgenic TCR expression.

True replacement of the endogenous TCR by a transgenic TCR would solve all of these problems at once. Advanced genetic engineering, e.g., using clustered regularly interspaced short palindromic repeats (CRISPR)/CRISPR-associated protein 9 (Cas9) [27,28,29], now makes such orthotopic TCR replacement (OTR) possible [26,30,31]. Excitingly, OTR may even significantly facilitate a leaner and more straightforward production process of clinical T cell products. Advantages and challenges of T-cell-based immunotherapies are summarized in Figure 1. In this review, we summarize the most recent advancements regarding OTR technology, discuss current shortcomings, and lay out future perspectives. Ultimately, OTR may provide the means to combine the safety, efficacy, and predictability of physiological T cells with the versatility advantages of engineering.

## 2. Role of the Remaining Endogenous TCR

TCR replacement indicates that the endogenous TCR is eliminated upon introduction of a transgenic TCR. Otherwise, remaining TCRs suppress surface expression and, therefore, diminish functionality through competition for a limited number of CD3 molecules [20,32]. Moreover, since the TCR is a heterodimeric receptor, transgenic α- and endogenous β-chains (and vice versa) can also mispair. Such mispaired (sometimes also called “mixed dimer”) TCR variants can lead to novel antigen specificities that are not negatively selected in the thymus. Mispairing was shown to lead to allo-reactivity in a human in vitro setting [22] and to GvHD or xeno-reactivity in a murine in vivo setting [21] by the groups of Heemskerk and Schumacher, but it has been fortunately, to date, not yet observed in any clinical setting. Nevertheless, the absence of such adverse events should be interpreted with caution since the number of patients treated with TCR engineered T cells is limited, and there seems to be general consensus in the field that mispairing has to be avoided for the sake of safety.

### 2.1. TCR Mispairing

Methods to prevent TCR mispairing were developed for nearly two decades, and they include endogenous TCR knock-down by RNA interference [33,34], murinization of TCR constant regions [35,36], TCR-specific disulfide bonds [37,38], single-chain TCRs [39], TCR domain swapping [40], co-delivery of accessory or co-stimulatory molecules [41], and TCR framework engineering [42]. All of these measures can enhance transgenic receptor expression, but do not completely eliminate mispairing. Therefore, genetic elimination of endogenous TCR loci was explored using new genomic engineering tools. In these cases, the constant regions of α- (T-cell receptor alpha constant, *TRAC*) and β-chain (T-cell receptor beta constant, *TRBC*) are typically targeted. In a seminal study, the group of Bonini could show that endogenous TCR knock-out (KO) through zinc finger nucleases combined with lentiviral transgenic TCR transduction leads to enhanced transgenic TCR functionality, as well as reduced allo- or xeno-reactivity [43]. Later, several other studies also provided proof of concept for endogenous TCR KO by CRISPR/Cas9 [44,45,46]. Of note, “endogenous TCR KO” in many studies refers to KO of the endogenous α- or β-chain alone, and this approach was suggested to be sufficient [46]. Our group could recently show that mispairing upon single-chain editing is highly TCR-dependent, which is in line with differential promiscuity of individual TCR chains [42], and only editing both endogenous α- and β-chains completely abrogates TCR mispairing [26].

### 2.2. Risks and Opportunities of Multiplexed Gene Editing

Single-chain editing can even drive TCRs into enhanced mispairing when, e.g., for a given endogenous β-chain only the transgenic α-chain is left as a pairing partner [26]. This would argue for dual α- and β-chain editing, but multiplexed editing (i.e., induction of double-strand breaks at more than one position) is not without disadvantages. It increases the number of reagents which need to be produced according to good manufacturing practice (GMP) and increases the risk of off-target effects, as well as the risk of chromosomal translocations [47]. Therefore, for the sake of clinical feasibility and safety, as few loci as possible should be edited. In a first in-human phase I clinical trial with CRISPR/Cas9-edited T cells specific for NY-ESO-1, June and colleagues targeted both *TRAC* and *TRBC*, in addition to Programmed Cell Death 1 (*PDCD1*) [47]. While no GvHD was observed, *TRAC* and *TRBC* KO was incomplete (KO efficiencies for TRAC and TRBC were around 45% and 15%, respectively), which can–as stated above–even increase the risk of TCR mispairing. Translocations through multiplexed TCR editing and *PDCD1* editing [48] have the potential to lead to malignant transformation of edited T cells, but such transformation was not observed in the clinical trial [47]. Instead, the authors argued that *PDCD1* editing only led to a desired effect, which is long-term maintenance of the edited cells [47].

### 2.3. Deliberate Conservation of Endogenous TCRs

In certain clinical settings, leaving the endogenous TCR untouched may even be desirable. In a clinical trial performed by the Greenberg group, Epstein–Barr virus (EBV)-specific T cells with unedited endogenous TCRs served as host T cells for a transgenic TCR specific for Wilms Tumor 1 (WT1) [16]. The authors argued that this should decrease the risk of GvHD through mispairing, presumably due to the limited number of potential TCR chain pairing partners, or through the fact that TCR chains that are specific for a foreign epitope may have a decreased likelihood of simultaneously bearing reactivity against self-epitopes. These considerations may be valid, even though the risk of mispairing would be only fully eliminated upon complete genetic KO of endogenous TCR chains. Leaving the endogenous TCR unedited also provides a much more intriguing opportunity, which is an in vivo vaccination effect mediated through, e.g., latent EBV reservoirs triggering the endogenous EBV-specific TCR, thereby leading to enhanced maintenance of the TCR-transgenic T cells [16]. Concerning CAR T cells, unaltered endogenous TCR expression is not problematic with regard to TCR mispairing, although GvHD may also be caused by the regularly paired endogenous TCR itself. Interestingly, a certain level of self-reactivity may also have a beneficial effect in terms of sustaining CAR T cell maintenance [49].

In summary, *TRAC* and *TRBC* dual KO can eliminate the risk of mispairing, but only when editing is complete since, otherwise, mispairing may even be increased [26]. Complete KO of the endogenous TCR may also improve T-cell functionality through increased surface expression of the transgenic TCR. However, multiplexed editing automatically introduces additional risks through off-target effects and chromosomal translocations. Editing of additional loci such as *PDCD1* to, e.g., enhance T-cell maintenance, likewise brings along caveats (such as malignant transformation). In certain clinical scenarios, leaving the endogenous TCR unedited may even be desirable. Overall, opportunities and risks through multiplexed editing need to be carefully weighed.

## 3. True TCR Replacement through Orthotopic Editing

Conventionally, TCR-transgenic T cells are generated through viral transduction, leading to untargeted insert integration into genomic DNA. Sleeping beauty transposon systems more effectively target so-called genomic safe harbor loci [50], but at best also lead to semi-random integration. In addition to the safety risks through uncontrolled editing of endogenous gene loci, these approaches make constitutively active, extrinsic gene promotors necessary to drive TCR transgene expression. In contrast to non-engineered T cells, TCR-transduced T cells do not show effective TCR downregulation after antigenic stimulus [25,26]. Furthermore, systematic comparisons of integration sites upon viral transduction and targeted transgene insertion using CRISPR/Cas9 are still lacking.

### 3.1. OTR Enables Engineering of Near-Physiological T Cells

In the form of DLI [5], TIL [6], or (virus) antigen-specific T cells [7], physiological T cells have proven their therapeutic value in the clinic for more than 25 years and, thus, shown excellent safety profiles. With the advent of genomic engineering possibilities through tools such as CRISPR/Cas9, we and others, therefore, set out to position transgenic antigen-specific receptors into the endogenous TCR gene locus [26,30,31]. By electroporation of guide RNA (gRNA)–Cas9 ribonucleoprotein (RNP) and a TCR DNA template [51], transgenic TCRs can be inserted into specific endogenous gene loci using homology-directed repair (HDR). Proof of concept for this was first provided by the Sadelain group using adeno-associated virus (AAV)-mediated delivery of a CAR DNA template [30], and later refined through the Marson group by making this process completely non-viral and also demonstrating feasibility for TCR engineering [31]. No extrinsic promotors are supplied. Instead, endogenous and transgenic TCR transcription and translation are linked via a self-cleaving peptide such as a P2A element (Figure 2). This should lead to both physiological regulation of the transgenic TCR and truncation of the endogenous TCR after peptide cleavage. We hypothesized this would generate TCR-transgenic T cells that are more similar to non-engineered T cells and, in particular, we were intrigued by the possibility to hit two birds with one stone, i.e., generation of T cells with physiological regulation of a transgenic TCR, as well as simultaneous elimination of the endogenous TCR. Technically, only such a combination of orthotopic TCR insertion in the endogenous TCR gene locus with elimination of the endogenous TCR locus reflects true TCR replacement (i.e., orthotopic TCR replacement, OTR).

### 3.2. TCR Regulation

OTR with transgenic TCRs and CARs leads to more dynamic downregulation of transgenic antigen-specific receptors after antigenic stimulation [26,30], in resemblance to non-engineered T cells. While these results clearly indicate that OTR T cells are differentially regulated compared to conventionally edited T cells, many aspects of how TCR regulation is different remain to be understood. TCR regulation was studied in detail at the protein level [52], but the role of genetic control in TCR expression is poorly characterized. A central reason for this is the lack of experimental tools to investigate constitutive versus dynamically regulated genetic transcription of the same TCR side by side. OTR now enables to address this issue through detailed analyses of the individual contribution of TCR gene and protein regulation (Figure 3). At the genetic level, it is also unclear whether TCR α- and β-chains, whose gene loci lie on different chromosomes, are differentially regulated. While most OTR protocols insert the full transgenic αβ TCR into endogenous TRAC (and merely knock out endogenous TRBC), we and others also provided proof of concept for the feasibility of individual knock-in of transgenic TCR α- and β-chains into their specific endogenous counterparts [26,31]. Whether this approach results in noticeably differential (potentially even more physiological) regulation is not yet addressed. Finally, current OTR protocols insert the new TCR into endogenous TCR constant regions, yet each TCR V segment has a unique promotor and leader sequence, and this may influence gene regulation of individual natural TCRs.

### 3.3. Defined Antigen Receptor Expression through OTR

Targeted TCR insertion not only allows placing an antigen receptor into a desired gene locus, but also brings along tight control over transgene copy numbers and expression levels (Figure 3). TCR and CAR expression are more defined after OTR than after viral transduction [26,30]. TCR expression levels have a significant impact on T-cell functionality [32,42,45,53,54,55]. In gene therapy protocols, transgene copy number is a key safety parameter and, for clinical T cell products—as for any medicinal product—predictability is crucial. In contrast to conventional editing methods, OTR, therefore, bears the potential to generate more predictable cell products. In this context, it would be particularly valuable if OTR could also decrease donor-to-donor variability between therapeutic cell products, which represents a major bottleneck for clinical application. Systematic investigation of this is missing.

A further important unresolved question is how often OTR editing occurs on only one and how often it occurs on both TCR gene locus alleles. Using a fluorochrome knock-in mix, Roth et al. demonstrated that bi-allelic insertions occur more often than expected by chance [31]. Further exploration of this issue will also enable us to address the question of how allele-specific TCR regulation is. For example, TCRs may be differentially regulated when RNA is transcribed from “second” α- or β-chains that did not undergo allelic exclusion. Recent work also demonstrated that TCR gene expression is also regulated by more distant control elements, such as HLA class II [56].

### 3.4. Functional Consequences of OTR

TCR downregulation after antigen contact most probably evolved as a negative feedback loop to protect T cells from excessive stimulation and to ensure long-term maintenance, in particular during chronic infections and cancer [23]. Whether or not OTR T cells perform “better” will likely depend on the clinical context (i.e., antigen load, kinetics of antigen presentation, affinity of TCR–antigen interaction). In short-term in vitro assays, OTR T cells performed comparably to conventionally TCR transduced T cells in terms of cytokine release or target cell killing [26,30,31]. In an intermediate-term in vitro “stress test” and in xenograft mouse models, OTR T cells outperformed conventionally edited T cells, and this was linked to decreased exhaustion and enhanced maintenance of a less differentiated phenotype [30,31]. However, in specific clinical scenarios, unphysiologically high TCR expression may be desirable, such as when antigen loads are low and priority is given to eliminating cells with residual antigen expression [16]. In other clinical scenarios, e.g., when antigen loads are high or when T-cell exhaustion quickly occurs, physiological TCR regulation may be absolutely crucial for optimal therapeutic efficacy.

Such a differential, context-dependent importance of TCR regulation patterns mirrors considerations on complementary and/or synergistic effects of T cells with different TCR–peptide major histocompatibility complex (pMHC) affinities [57,58]. Physiological T-cell responses not only consist of T cells with dynamic TCR regulation, but they are also polyclonal. TCR–pMHC affinity was shown to impact TCR downregulation in individual clones. In a study by the van Heijst group, CD4^+^ T cells with high TCR affinity underwent a higher degree of TCR downregulation compared to T cells with low TCR affinity [24]. The authors discussed that such “programmed TCR downregulation” could represent an evolutionary solution to tailor T-cell responses in a way that balances robustness of effector function with avoidance of immunopathology. As intriguing as this hypothesis is, the TCRs with differential affinities used in these studies were derived from TCR transgenic mice. Despite the fact that these T cells do show substantial TCR downregulation, it is questionable how physiological TCR regulation in these cells is. Generally, long-term aspects such as T-cell exhaustion and maintenance are best evaluated in syngeneic mouse models, for which proof of concept of OTR is not yet provided. Closing this missing gap represents one of the most important next steps for the field of TCR engineering.

### 3.5. OTR Mouse Models

TCR transgenic mice proved invaluable for research addressing both basic immunology and T-cell therapy. Over the past three decades, sufficient cell numbers of undifferentiated TCR transgenic cells present in the periphery of such mice provided a robust and standardized source of CD4^+^ or CD8^+^ T cells for a sheer endless number of in vitro and in vivo experiments worldwide. TCR transgenic mice with specificities for HY (HY TCR [59]), ovalbumin (OVA) (OT-I [60] and OT-II [61] TCR), glycoprotein (gp) (P14 [62] and SMARTA [63] TCR), or pigeon cytochrome C (5C.C7 TCR [64]), therefore, profoundly shaped our understanding of T-cell immunity.

TCR gene expression in TCR transgenic mice is not physiological when heterologous gene promotors are used, and, for this reason, in some TCR transgenic mice, the genomic DNA of whole TCR gene loci was integrated, in order to include as many natural regulatory elements as possible [61]. However, regulatory elements can be positioned very far from the coding gene itself. Furthermore, most of the TCR transgenic mice that are widely used are often highly heterogenous and unphysiological in terms of their TCR genetics. For example, the OT-I TCR α-chain offered by the Jackson Laboratory (#003831-JAX) is expressed under control of the H-2k^b^ promotor, a fragment of an immunoglobulin H (IgH) chain enhancer, as well as other fragmented non-coding DNA sequences. The P14 mouse offered by the Jackson Laboratory (#37394-JAX) is estimated to harbor 10–20 TCR copies at an unknown location in the genome.

In light of this, it will be exciting to study whether experiments using conventional TCR transgenic mice were subject to a systematic bias or whether, due to the singularity of the insertion site and the genomic context of inbred mice, individual TCR transgenic mice show substantial differences in TCR regulation. Such differences could dramatically affect T-cell function, particularly in mouse models investigating chronic immune responses. Investigation of these questions relies on generation of OTR transgenic mice. This in turn requires targeted integration of large genes (>2 kb) such as the TCR into the genome of mouse zygotes. Unfortunately, despite considerable progress in the field of CRISPR/Cas9-mediated mouse zygote editing, such editing is yet to be achieved with reasonable efficiency [65,66,67,68,69,70,71,72,73].

On an interesting note, systematic comparisons of TCR regulation in TCR transgenic versus TCR “retrogenic mice” [74] would be informative, and they have been technically possible for a long time; however, to our knowledge, such studies have not been undertaken or reported. TCR retrogenic mice are generated via retroviral transduction of hematopoietic stem cells, which are used to create bone marrow chimera. Mature T cells in these mice express the “retrogenic TCR” from various integration sites under a constitutively active promotor, which should contrast with genetic TCR regulation in T cells from conventional transgenic mice.

Overall, given the wide use and importance of TCR transgenic mouse models, our lack of knowledge regarding precise TCR regulation in these models is surprising. With further technological development in the field of mouse zygote editing, investigation of genetic TCR regulation in conventional TCR transgenic mouse models will be of fundamental significance.

### 3.6. Defined TCR Insertion outside the Endogenous TCR Locus

Upon TCR transduction, the TCR is semi-randomly integrated into the genome. It will be interesting and important to understand how TCR transcription is regulated in gene loci other than the endogenous TCR locus, as investigation of this in TCR-transduced T cells will inform us regarding TCR regulation in cells that have been widely used in immunological research for decades. It is also noteworthy that off-target integration of antigen-specific receptors led to spectacular, although unintended, findings on altered cellular function through genome editing. The most prominent example is the observation that integration of a CAR into the Tet Methylcytosine Dioxygenase 2 (TET2) locus led to presumed “positive selection”, providing in-human proof of concept that single engineered T-cell progeny can, under the right circumstances, yield protective immunity [75]. Generally, however, (semi-)random TCR insertion seems an undesirable option for future clinical application given the safety hazards.

The ability to position transgenes in any desired gene locus in a highly precise manner opens up many new avenues of possibilities. Recently, insertion of CAR transgenes into the endogenous gene loci of interleukin-2 receptor subunit alpha (*IL2Rα*) or *PDCD1* gene loci was proposed to generate “smart CAR T cells” which act in accordance with phenotypic differentiation kinetics [76]. It is uncertain, however, how often the scientific community will be able to “outsmart” evolution. Instead, given the influence of infectious diseases on the evolution of our immune system [77], it appears that T cells evolved in a way that makes them highly efficient to combat infections. Irrespective of one’s personal view on these considerations, novel genomic engineering tools such as CRISPR/Cas9 enable us to design T cells in a directed and precise manner. Further technological refinement of methods such as OTR is critical to make these engineering possibilities a clinical reality for the benefit of patients.

## 4. Technology Development

### 4.1. gRNA Cas9 Ribonucleoproteins

A breakthrough for applying CRISPR/Cas9-mediated editing to primary human T cells was the development of protocols with electroporation of gRNA–Cas9 ribonucleoproteins (RNPs) by Schumann et al. [51]. RNPs lead to more efficient editing compared to gRNA and/or Cas9 in the form of DNA, since gRNA and Cas9 protein are directly available in their needed form. At the same time, degradation of RNPs [78] and a lack of genomic integration make RNPs the safer option, since off-target effects through constitutive editing are reduced [79]. RNPs also circumvent the problem of DNA toxicity to the cells (compared to delivery of gRNA and/or Cas9 via, e.g., plasmid DNA), and they seem to reduce toxicity of donor template DNA [31]. Off-target effects can be further reduced by the use of high-fidelity Cas9 variants [80,81], in the case of newer variants often even without loss of on-target efficacy [79].

### 4.2. Off-Target Effects

The specificity of CRISPR/Cas9 editing is a continuous focus of research, and the scientific community is still in a learning process with regard to finding sensitive technologies to detect off-target effects, as well as to finding meaningful ways of data interpretation. In analogy to Heisenberg’s uncertainty principle [82], detecting off-target effects underlie an inherent limit of precision with which off-targets can be detected in a sensitive and, at the same time, broad manner. Either predicted off-target sites can be investigated in great depth at the cost of narrowing down screenings to a limited number of sites (and with the uncertainty of the fidelity of off-target site predictions) or off-target effects can be investigated on a genome-wide scale at the cost of sensitivity to detect individual off-target events. Methods to detect and reduce off-target effects induced by CRISPR/Cas9 were extensively reviewed elsewhere [72]. In the case of OTR, three types of off-target effects are particularly relevant: (1) off-target double strand breaks through the Cas9 nuclease, (2) off-target integration of donor template DNA, and (3) chromosomal translocations through multiplexed editing.

As indicated above, the combination of RNPs and high-fidelity Cas9 variants significantly reduces the frequency of off-target effects. Behlke, Porteus, and colleagues reported the frequency of off-target events to be 28–72% with constitutive Cas9 expression, which is reduced to 3–12% with RNPs and further reduced to 1% with high-fidelity Cas9 RNPs [79].

Off-target integration of donor template DNA into the genome is more frequent into off-target double-strand breaks than into intact off-target sites. Applying the OTR protocol with RNPs, Roth et al. reported frequencies up to 1% for the former and frequencies of around 0.1% for the latter when looking at single-cell GFP expression via flow cytometry after provision of off-target gRNAs or no gRNAs, respectively. These values could be further reduced to 0.01% (around the limit of detection for this assay) off-target integration when single-stranded DNA (ssDNA) instead of double-stranded (dsDNA) was used for the donor template DNA [31]. These results were corroborated by targeted locus amplification (TLA) [83] on a genome-wide scale, for which the limit of detection is about 1%. Our own group could confirm these results using OTR through *TRAC* and *TRBC* editing with RNPs. While we saw low-frequency (about 0.1–1%) integration into *TRBC* by both flow cytometry and TLA when *TRAC*-targeting HDR constructs were used, we could not detect any integration outside loci with intentionally induced double-strand breaks [26].

We already outlined above that multiplexed editing can lead to chromosomal translocations. In their first in-human phase I clinical trial with CRISPR/Cas9-engineered TCR-transduced human T cells, Stadtmauer et al. reported translocation frequencies of about 0.01% to 5% [47], which is in line with previous reports on translocation frequencies of Transcription activator-like effector nuclease (TALEN)- [84,85] and CRISPR-edited [86] T cells. Translocation frequencies are different for individual gRNA combinations [47,86] and can, therefore, be reduced through informed selection of specific gRNAs. In addition, Cas9 variants such as base editors that circumvent double-strand breaks were shown to decrease translocation events significantly [86].

While these advances are applaudable, what do these numbers actually mean? As for any medicinal product, the risk–benefit ratio of gene and cell therapies will be very dependent on the specific clinical scenario, i.e., the underlying disease, treatment alternatives, or the cell type used for editing [87]. In T cells, translocations most often do not result in malignant transformation and do not exclude the possibility of long-term persistence [47]. As of 2012 [88], and, to the best of our knowledge, still today, no genetically engineered T cell was ever reported to undergo malignant transformation. Most of these therapeutically used cells were transduced with a CAR without further gene editing, e.g., through CRISPR/Cas9. Although, hypothetically, off-target effects through viral transduction (as through semi-random transgene insertion) should be expected to occur more frequently than off-target effects through CRISPR/Cas9 editing, this was never actually tested in a systematic manner. Overall, we expect OTR T cells in terms of off-target effects to be sufficiently safe for most clinical applications. Actual clinical data supporting this claim are urgently needed.

### 4.3. Non-Viral DNA Template Delivery

Initial proof of concept for OTR by a CAR was provided through AAV-mediated delivery of the CAR DNA template [30]. AAVs (in particular, serotype 6) are being successfully employed in various gene therapy approaches, especially for in vivo targeting of liver, eye, or hematopoietic stem cells [87]. While delivery in these settings is usually the rate-limiting factor and AAV-mediated delivery is, therefore, a crucial component, T cells can be manipulated ex vivo in more flexible ways. Non-viral OTR is possible via electroporation of transgenic receptor DNA in the form of linearized DNA (ssDNA or dsDNA) or plasmid DNA [26,31]. Non-viral editing is, thus, less efficient, but entails crucial advantages for clinical application. A completely non-viral editing process is more likely to be approved by regulatory authorities. Furthermore, and perhaps even more importantly, non-viral delivery can be easily scaled up to editing of many different receptors at the same time. The target heterogeneity of tumors and the HLA restriction of tumor as well as pathogen epitopes, are—in nature—met by an equally diverse TCR repertoire landscape. Targeting different epitopes decreases the likelihood of tumor or pathogen escape, and targeting the same epitope with different receptor affinities brings along natural advantages of synergy and complementarity [57,89,90]. Finally, interindividual differences between patients require T-cell-based therapies to be often highly personalized.

### 4.4. DNA Template Design

The most commonly used OTR protocol is performed with a DNA template design that directs the full αβ TCR into endogenous *TRAC* with concomitant *TRBC* KO (Figure 2). The gRNA cutting and, therefore, template insertion site is positioned at the beginning of the first exon of TRAC. After the left homology arm, the first fragment of the insert is a P2A element, positioned in frame with endogenous TRAC. After natural V(D)J transcription and translation, this self-cleaving peptide element leads to cleavage of the transgenic TCR from the endogenous V(D)J segments. The P2A element is followed by the TCR β-chain, another 2A element, and the TCR α-chain. When the transgenic TCR employs human constant regions, the insert only needs to contain the sequence of the first exon of TRAC which is positioned 5′ of the Cas9 cutting site. From then on, the transgene sequence seamlessly continues into the endogenous TRAC locus with the right homology arm. Although, with this DNA template design, OTR robustly works, the significance of many technical details is still unclear.

#### 4.4.1. Gene Locus

Current OTR protocols target TCR constant regions. However, and as discussed above, each V segment has its individual promotor. The significance of this is unexplored. Furthermore, it is not clear whether the truncated endogenous V(D)J segments play any role when constant regions are targeted.

#### 4.4.2. Homology Arms

Homology arms are 300–400 bp in length, with actual inserts being about 1.3–2 kb long. In general, short homology arms facilitate an easier and cheaper production process, but they may endanger the specificity of insertion. Extensive testing is ongoing in the genome editing field, whose homology arms can yield the best efficiency and specificity, including, e.g., testing of asymmetric homology arms [91] or short homology arms with DNA modifications [92].

#### 4.4.3. Cutting Site

Due to the triplet code of DNA, there is a 2/3 chance that cutting sites induced by Cas9 are out of frame. There is convincing evidence that the distance between the cutting site and the insert should be as small as possible [93], but it is unclear whether cutting sites that are directly in frame are disproportionately superior to short-distance cutting sites, or whether up- and downstream distances have a similar impact on insertion efficiency and specificity.

#### 4.4.4. Splice Donor/Acceptor

Some OTR DNA templates include splice acceptors [30] to enhance splicing of endogenous V(D)J to the constant region harboring the transgenic receptor, whereas others do not [26,31]. Whether this matters is unclear.

#### 4.4.5. Transgenic TCR Constant Regions

Modifications of TCR constant regions such as murinization [35,36] and inclusion of cysteine bridges [37,38] were mainly developed to prevent TCR mispairing, but they often also generally enhance TCR surface expression even in the absence of the endogenous TCR (own unpublished observations) and provide a practical tracking marker (e.g., via antibody staining for the murine constant region). We could also show, however, that human constant regions lead to more physiological dependencies of TCR expression on CD3 [26]. These benefits of enhanced vs. physiological TCR expression will need to be weighed against each other depending on the application. When murine constant regions are used, the full constant region needs to be provided, whereas, with human constant regions, the sequence can seamlessly continue into the endogenous TRAC locus. In our direct comparisons we also provided the full human TCR α-chain constant region within the insert, including a poly A tail after the end of the constant region. How such modifications further impact physiological engineering remains to be addressed in sufficient detail.

### 4.5. HDR Enhancement

HDR is, compared to non-homologous end joining (NHEJ), a naturally inefficient mechanism. Homology-independent targeted integration (HITI) was probed as a strategy to improve knock-in efficiencies [94]; however, to our knowledge, this was not conclusively or successfully tested in primary T cells. Small molecules that act as inhibitors of NHEJ or enhancers of HDR were also reported to lead to higher insertion efficiencies in T cells [95]; however, in the published report, overall insertion efficiencies in T cells were rather modest and, in our hands, such molecules did not lead to improved editing efficiencies (at least when viability of cells was also taken into account; unpublished data).

The above-mentioned technological developments represent some of the most relevant aspects of editing, but they are certainly not exhaustively listed. Moreover, we focused on CRISPR/Cas9 and we did not touch on new advances with other nucleases. Despite the fact that there is still a lot of room for technological improvement, independent and conclusive proof of concept for the feasibility of CRISPR/Cas9-mediated OTR in primary T cells was provided [26,30,31], and it is currently being explored as a GMP-conform editing method to generate antigen-specific transgenic T cells worldwide.

## 5. Toward Clinical Application of OTR

Development of OTR technology toward clinical application will require advances on many different levels. In this review, we focus on protocol translation as a GMP process, cellular phenotype, and activation for optimal in vivo efficacy, as well as required cell numbers and suitable cell selection processes.

### 5.1. GMP Process

Clinical OTR technology will take current clinical protocols, e.g., for production of CAR transgenic T cells by means of viral transduction, as a basis. The most notable differences concern the mode of delivery (i.e., electroporation), the use of CRISPR/Cas9 reagents, and the design of targeting constructs. Electroporation devices enable both editing in closed-circuit systems and multiplexing. Transparency of electroporation settings, as well as buffers [96], is an important aspect for GMP production. The use of Cas9–gRNA RNPs facilitates clinical-grade production since reagents can be produced recombinantly in a straightforward manner. As mentioned above, the rapid degradation of RNPs inside the cells represents a significant safety advantage [78]. Likewise, completely non-viral production processes (including the targeting construct delivery) facilitate regulatory approval. Systematic comparisons of non-viral (electroporation) vs. viral editing processes with regard to safety, viability, costs, throughput/upscaling, and/or reliability (list non-exhaustive) are important next steps to be taken.

### 5.2. Cell Phenotype and Activation

Naïve (T_N_), memory stem (T_SCM_), or central memory (T_CM_) T cells represent phenotypic subsets that are associated with T-cell maintenance and, thus, protectivity [97,98,99]. Interestingly, the above-mentioned TET2 editing led to preservation of a T_CM_ phenotype, potentially providing an important mechanistic explanation for the contribution of this individual clone to the overall T-cell response [75]. In a seminal study by the Riddell group using non-human primates, T-cell clones derived from different phenotypic subsets ex vivo adapted analogous phenotypes during in vitro culture, but subsequent T-cell maintenance in vivo in secondary recipients after adoptive transfer still correlated with the initial ex vivo phenotype [100]. This was later confirmed with different subsets of CD8 and CD4 CAR transduced T cells in mice [101]. However, it seems that the in vitro induced phenotype also matters to some extent, since more differentiated cells are less well maintained after adoptive transfer than less differentiated cells that stem from the same phenotypic precursors, as shown by Luca Gattinoni and Nicholas Restifo [102]. Systematic analyses of whether the initial ex vivo or the in vitro adapted phenotype is more important for later in vivo performance are lacking, but overall there is general consensus about the proliferative and, thus, protective capacity of less differentiated T cells. Therefore, it will be important to examine how well different T-cell subsets can be genetically engineered via OTR, and how well a stem-like phenotype can be preserved during the OTR protocol. So far only preliminary data exist that specifically addressed how well different T-cell subsets can be electroporated [103], and direct comparisons of phenotypic changes during OTR vs., e.g., antigen–receptor transduction are completely missing.

Importantly, short production and expansion processes are linked with conservation of less differentiated phenotypes in vitro and, thus, better in vivo performance. One of the few studies that investigated this systematically showed that CAR T cells harvested at day 3 showed better tumor control than T cells harvested at day 9, despite a six-fold lower dose of absolute cells [104]. The straightforwardness of the OTR protocol makes it seem possible to reduce the manufacturing time of genetically engineered T cells to an absolute minimum.

### 5.3. Cell Numbers and Selection

The lower efficacy of engineering T cells via OTR in a non-viral manner compared to AAV-mediated DNA template delivery or conventional transduction may pose a concern regarding implementation into clinical protocols. However, firstly, OTR cells can be easily selected with high purity and expanded [26]. Secondly, when T cells harbor a phenotype with high proliferative capacity, extremely low cell numbers can be sufficient for robust protection—sometimes even a single T cell can be enough [9,75,105,106,107,108,109]. Thirdly, low cell numbers are particularly compatible with robust functionality when they are selected with minimal manipulation, e.g., through reagents such as antigen-binding fragments (Fabs) [110,111] or peptide MHC streptamers [9,112]. Fourthly, transfection for OTR can be easily scaled currently with electroporation of, e.g., 1 × 10^9^ cells in a single tube.

For enhanced selection, OTR template designs can also induce additional markers such as a strep-tag [113,114]. Strep-tags can also be used for T-cell expansion and in vivo monitoring [115]. Similarly, a truncated version of the epidermal growth factor receptor (EGFRt) can not only be used for T-cell tracking, but also serves as a target for elimination of transgenic T cells via administration of the clinically approved antibody cetuximab [116], thereby representing an important module for engineering safe T-cell products. OTR template constructs with the same TCR, but different additional tags (e.g., strep-tag or EGFRt), can also be combined. Dual strep-tag- and EGFRt-expressing OTR T cells then indicate bi-allelic editing (unpublished data) and may represent particularly safe and simultaneously versatile T-cell products. Overall, advanced engineering can be used for generation of transgenic T cells that reflect physiological T cells more closely than ever before (“physiological engineering”), while it can also be used for defined introduction of selection markers and/or safety switches (“para-physiological engineering”).

## 6. Concluding Remarks

The success of immunotherapies, whether checkpoint inhibitors or CAR T-cell therapies, on the one hand, highlights that using natural defense mechanisms can be immensely effective to fight tumors or infections, but the necessity to supply these therapies in the first place and the fact that we need to modulate the immune system, on the other hand, also show that trusting in natural systems alone is sometimes not enough. In other words, we may be wasteful if we do not make use of physiological mechanisms that evolved over millions of years and, in most cases, provide robust immunity, but we will also need to tweak systems just a little bit at the right time in the right place in order to convey optimal protection. To this end, we regard OTR/physiological T-cell engineering as a baseline editing process that allows generating highly defined T-cell products which can be—if necessary—specifically modified to be most suitable for a specific clinical setting.

Physiological immune responses are not only dynamically regulated (e.g., through regulation of the TCR), but also diverse in terms of their composition. At the T-cell level, multiple TCRs targeting different epitopes or the same epitope with different affinities form a multi-faceted T-cell response. CD4^+^ and CD8^+^ T cells interact and convey synergistic and complementary functions. Zooming out even further, B and T cells together provide adaptive immunity through humoral and cellular effector function. Analogously to OTR, proof of concept of B-cell receptor (BCR) exchange was recently provided [117,118,119,120]. The versatility of novel gene editing tools, therefore, creates prospects for the feasibility to engineer complex composite immune responses that reflect natural immunity as much as possible while providing additional or altered function as much as necessary.

## Figures and Tables

**Figure 1 cells-09-01367-f001:**
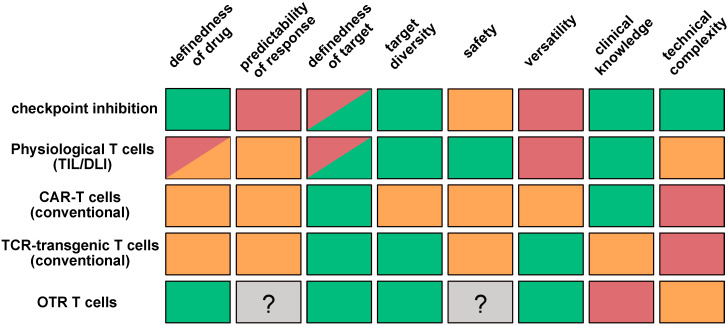
Advantages and challenges of T-cell-based immunotherapies. Checkpoint inhibition, non-engineered T cells, chimeric antigen receptor (CAR) T cells, T-cell receptor (TCR) transgenic T cells, and orthotopic TCR replacement (OTR) T cells differ in the fundamental characteristics of therapies. Colors indicate the degree of experience, benefit, or feasibility depending on the category (green: advantageous, yellow: medium, red: disadvantageous, gray: unknown). Given its novelty, clinical knowledge is limited for OTR T cells. OTR does, however, bear the potential to recombine many of the individual advantages of alternative strategies.

**Figure 2 cells-09-01367-f002:**
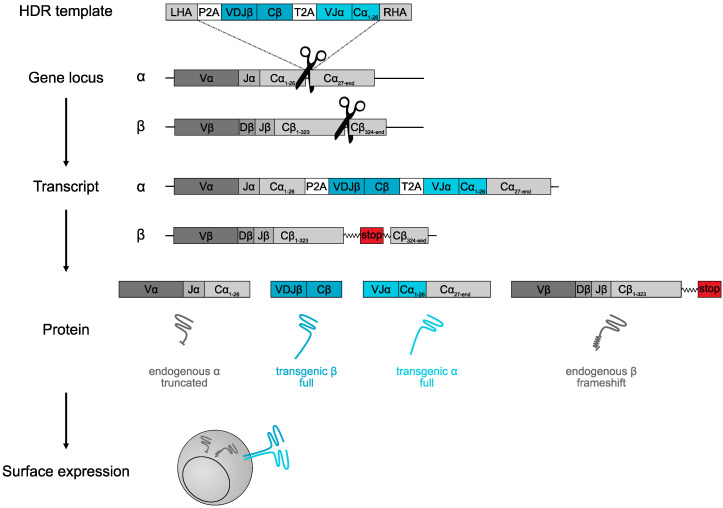
Schematic of orthotopic TCR replacement through clustered regularly interspaced short palindromic repeats (CRISPR)/CRISPR-associated protein 9 (Cas9)-mediated gene editing. The shown homology-directed repair (HDR) template is designed in a manner that the complete αβ TCR is integrated into the endogenous T-cell receptor alpha constant (*TRAC*) gene locus upon a CRISPR/Cas9-mediated double-strand break. Simultaneously, T-cell receptor beta constant (*TRBC*) is edited, and non-homologous end joining leads to knock-out of the endogenous TCR β-chain. Cleavage of 2A elements leads to intracellular expression of transgenic TCR chains and a truncated endogenous TCR α-chain; thus, at the surface, only the transgenic αβ TCR is expressed. LHA, left homology arm. RHA, right homology arm. P2A, T2A, self-cleaving peptides. VJ/VDJ, variable parts of TCR. Cα, TRAC. Cβ, TRBC. For more information, see References [26,31].

**Figure 3 cells-09-01367-f003:**
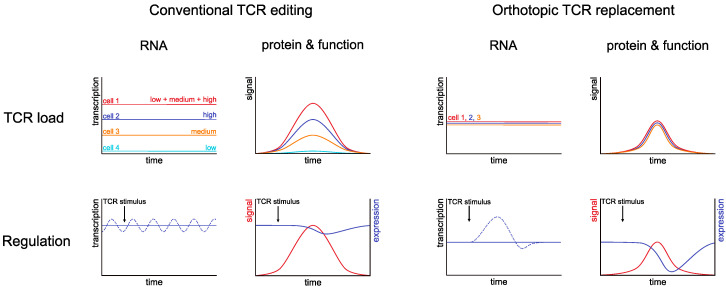
TCR RNA and protein levels and their impact on function after conventional editing versus OTR. Top left: Conventional TCR editing leads to random insertion of TCR DNA and, thus, expression of TCR RNA. Individual cells can harbor high or low expression of individual transcripts, or a combination of them, depending on the site of insertion and the copy number. This directly affects protein expression and TCR function. Bottom left: Upon insertion in non-TCR gene loci, baseline TCR transcript regulation may be more unstable (and, e.g., affected by “global” or “off-target” genetic regulation of the cell). Furthermore, regulation should not be affected by TCR stimulation, leading only to minor shifts in TCR protein expression (as experimentally shown in References [25,26]). Top right: Upon OTR, transgenic TCR insertion is highly defined, leading to defined protein expression and cellular function. Bottom right: TCR RNA expression should be more affected by TCR stimulation after OTR. At the protein level, the initial more significant downregulation of TCR protein (as experimentally shown in Reference [26]) is followed by upregulation. While first experimental evidence exists for differential TCR regulation upon OTR, the exact contribution of TCR insertion (copy number and integration site), RNA expression, and protein regulation will need to be delineated in detailed studies in the future.

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
