# Peer review of "Orthotopic T-Cell Receptor Replacement—An “Enabler” for TCR-Based Therapies"

_cells, 2020, doi:10.3390/cells9061367_

Round 1

Reviewer 1 Report

In the review manuscript entitled „Orthotopic T cell receptor – an ´enabler´ for TCR-based therapies, Schober and co-workers have critically summarized the knowledge about some of the current available methods for T-cell based therapies. In particular they focused on orthotopic TCR replacement by CRISPR/Cas9 technologies.

The review is interesting, well-structured and up-to-dated. However, the complexity of the phrasing is the major problem of this work. Sentences are in many cases long and grammatically intricate. Words are in some cases not used appropriately. This makes difficult for the reader to understand what the authors mean.

Author Response

We are glad the reviewer finds our review interesting, well structured and up-to-date. We acknowledge that some of the sentences have been too long and had the manuscript re-checked by an English native speaker (Mrs. Steffeni Mountford, see tracked changes in Word document). We have now edited several passages in order to enhance language clarity.

Reviewer 2 Report

The manuscript provides an excellent overview of the current clinically established as well as more experimental methods of TCR editing, concentrating and explaining in detail OTR replacement. It is a great contribution to the literature considering the authors latest publication (Nature Biomedical Engineering, 2019) and expertise in the field.  

The reviewer has been impressed with how comprehensive and clear in its writing the authors have been. 

Overall the reviewer only has minor comments in terms of editing the text and introducing minor charges:

1. Could the authors correct a typo in the abstract section ('lymphoctes') as indicated 

2. If possible, could authors rephrase some colloquial expressions as indicated in the PDF copy attached.

3. Is it fair to show unknown (indicated with question mark) effects of OTR editing in green in figure 1, suggesting they are advantageous even though we have no data available so far? Could authors consider alternative solution i.e. grey/ neutral background?

4. Reviewer indicated in some parts of the text the missing references.

5. Reviewer would suggest including and discussing briefly also beneficial (though unplanned) effects of off-target integration as described in reference 103 for Tet2 gene (off- targets effect section).

6. In the section for GMP production it would be important to comment/ compare the effect of viral vs electroporation gene delivery in terms of cells viability.

7. In the section for cell phenotype and activation, reviewer suggests including Freitta et. al. paper from 2018 providing a context for clinical responses in relation to CAR-T cells phenotype.

8. For the section on cell numbers and selection can authors comment (and provide a reference) if sorting has previously been successfully incorporated into GMP selection protocols.

9. In the last section, the reviewer would advise the rephrase the sentence using a word 'foolish', which to some readers can indicate a biased view.

Author Response

We thank the reviewer for the positive evaluation of our manuscript and for the constructive feedback. Regarding the specific points, we here provide direct replies:

  1. We corrected the spelling mistake, thank you for pointing this out.
  2. We have carefully re-read the manuscript with regards to colloquial language. While it is clear that colloquial language should be used in absolute exceptions, we stuck with a few expressions that are figures of speech, as should be legitimate in a review article.
  3. This is a good point. We have changed the background of the boxes with question marks to grey.
  4. We provided these references (references 30 and 31 at the beginning of page 5, reference 76 at the end of page 8, reference 99 at the end of page 12).
  5. We included a brief discussion of the TET2 KO effect in this part of the manuscript (2nd half of page 8 and middle of page 12).
  6. We included a statement that it will be important to systematically investigate viability and other aspects in direct comparisons between non-viral and viral editing methods (first part of page 12).
  7. We mentioned the two Fraietta et al. studies (the already cited one and the one proposed by the reviewer) in this context as well (middle of page 12).
  8. We have changed the wording of "sorting" to "selection" in this paragraph, as "selection" encompasses a broader field of techniques which have been clinically validated (and for which we have given examples also in the original version of the manuscript) (end of page 12).
  9. We have re-phrased this sentence (middle of page 13).

Reviewer 3 Report

Schober and colleagues describe the OTR technology that they published last year in the context of other available knock out-based strategies. They extend their discussion to its eventual clinical development. The review is comprehensive and covers the TCR development field. In my opinion, it is ready to be published, but I would recommend the following modifications:

  • Figure 1 does not look so nice (yellow appears orange on my printed version), the design is not very appealing, and the content is questionable. Maybe to focus on the advantages of OTR Tc would help?
  • p.11: Splice donor/acceptor: one or two explanatory sentences would help the ignorant reader to understand what this point is about 
  • The concluding remark should be modified (p.13): sentences such as "trusting in nature alone" or "have evolved over millions of years to protect us" are very difficult to defend in the context of a scientific article. One should avoid the usage of the word "nature"  and rather write "natural system" or "evolution". For the second sentence, the authors here open a rather controversial subject, do they have a teleological view of evolution? In the text (p.8) they talk about outsmarting evolution and cite a paper (cit 73). However this paper does not discuss how T cells "have evolved" but how infections have shaped our immune system, suggesting a mechanism with no apparent goals. 

Author Response

We thank the reviewer for the positive evaluation of our manuscript. The specific points raised by the reviewer are valid, and we have addressed them in a revised version of the manuscript.

  • We acknowledge that the design of Fig. 1 has not been optimal. We have now changed the colors to be less prominent, created a more balanced weighting of titles and boxes, and have changed the background of the boxes with question marks to a neutral grey. We do think that the inclusion of all kinds of treatment strategies helps to provide a broad context for physiologically engineered T cells.
  • We have included an explanatory sentence regarding splice acceptors (first part of page 11).
  • We have modified the concluding remarks (middle of page 13) and the previous section in the text (end of page 8) regarding the evolution of the immune system. We do not hold a teleological view on evolution. The reference to teleology was meant as a rhetorical trick to make clear that the result of evolution is an immune system that mostly works as robustly as if it had been planned. We have now a) deleted the words "teleologically appealing" in the text in order to avoid confusion, b) cited previous reference 73 more carefully, c) re-phrased this section to enhance clarity that evolution did not take place with a goal, but has nonetheless led to T cells being very efficient in what they do, d) exchanged the word "nature" by "natural mechanism" in the concluding remarks as requested, and e) exchanged "protect us" by "in most cases provide robust immunity" as requested.

Reviewer 4 Report

The manuscript from Schober et al is a review of their recently published method of orthotopic TCR  replacement (OTR) for TCR therapies.  The review is comprehensive and clear and discusses the main methods/technologies available for TCR-based (and to some extent CAR) T-cell therapy for clinical trials with their advantages and disadvantages. Non-viral delivery of TCR or CAR has clear advantages for clinical use.

There is a lot of focus on TCR mispairing in the review as avoiding this may be one of the proven main advantages of replacing the endogenous TCR. However, TCR mispairing has, as the authors mention, never been seen in clinical trials so far. Despite that this possibility cannot be excluded, the number of patients that have so far received TCR therapy is not so limited; a recent meta-analysis of TCR-based therapy (PMID: 30798772) showed that 84 clinical trials were found (2018). Sixteen percent of these 216 cancers targeted were melanoma which accounted for 16% of the cancers with close to 200 patients for melanoma alone in a recent review of trials on clinicaltrials.gov. However, it is mcuh fewer than for CAR-based therapy.

The review is well written and, however, there are a few minor points:

1.       The authors repeatedly use the term physiological T cells. Despite the fact that a modified T cell can be said to have a TCR that is not proved to function normally or one that is non-appropriate to the cell, this use of the term may be misleading. The authors should rather use the term physiological when describing functions of the cell or TCR as in the article they previously published: “Orthotopic replacement of T-cell receptor α- and β-chains with preservation of near-physiological T-cell function”.

2.       Fig. 1 Visualizing advantages and disadvantages of T cell based immunotherapies could be improved. The squares are very dominant whereas the titles are hard to see. A table summarizing the different approaches and advantages on one hand and disadvantages on the other may be easier to follow.

3.       In the paragraph discussing TCR insertion outside the endogenous TCR locus, the authors mention in the 5th line that “….the community will be able to outsmart evolution and it is teleologically appealing that physiological T cells have evolved in a manner that renders them highly efficient to combat infections”. Whether there is a teleological explanation for T cell function is debatable. The fact that the immune system in each individual reacts to infections and adapts to them could be regardless of function or goal, but certain traits are passed on simply due to natural selection. The cited review on impact of migration and evolution on immunity does not mention teleology, but selective pressure.

A couple of typos/grammatical errors that should be corrected:

In figure 2, 2nd top left should say Gene locus

3rd line under HDR enhancement; sentence contains ”been” twice

7th line under Cell phenotype and activation contains “also” twice

Author Response

We thank the reviewer for the positive evaluation of our manuscript and the constructive feedback. We reply to the specific points raised as follows:

  • We have included the meta-analysis on TCR-engineered therapies in our manuscript, thank you for pointing this out (new reference 17).
  • We acknowledge the term 'physiological T cell' can be misleading when compared to TCR-transgenic T cells, which are also in many aspects still physiological even if TCR expression is unphysiological. In these direct comparisons, we have now more often used the term 'non-engineered' instead of 'physiological' (see tracked changes in the Word document).
  • We acknowledge that the design of Fig. 1 has not been optimal. We have now changed the colors to be less prominent, created a more balanced weighting of titles and boxes, and have changed the background of the boxes with question marks to a neutral grey. We do think that the inclusion of all kinds of treatment strategies and the respective categories helps to provide a broad context for physiologically engineered T cells.
  • We have modified the concluding remarks (middle of page 13) and the section in the text (end of page 8) regarding the evolution of the immune system. We do not hold a teleological view on evolution. The reference to teleology was meant as a rhetorical trick to make clear that the result of evolution is an immune system that mostly works as robustly as if it had been planned. We have now a) deleted the words "teleologically appealing" in the text in order to avoid confusion, b) referenced previous reference 73 more carefully, c) re-phrased this section to enhance clarity that evolution did not take place with a goal, but has nonetheless led to T cells being very efficient in what they do, d) exchanged the word "nature" by "natural mechanism" in the concluding remarks as requested, and e) exchanged "protect us" by "in most cases provide robust immunity" as requested.
  • We have corrected the spelling mistakes, thanks for pointing them out (Fig. 2, end of page 11, middle of page 12).

Round 2

Reviewer 1 Report

The manuscript has been significantly improved.